# Analyzing Modular Approaches for Visual Question Decomposition

**Apoorv Khandelwal** and **Ellie Pavlick** and **Chen Sun**

Brown University

Department of Computer Science

{apoorvkh,ellie_pavlick,chensun}@brown.edu

## Abstract

Modular neural networks without additional training have recently been shown to surpass end-to-end neural networks on challenging vision–language tasks. The latest such methods simultaneously introduce LLM-based code generation to build programs and a number of skill-specific, task-oriented modules to execute them. In this paper, we focus on ViperGPT and ask where its additional performance comes from and how much is due to the (state-of-art, end-to-end) BLIP-2 model it subsumes vs. additional symbolic components. To do so, we conduct a controlled study (comparing end-to-end, modular, and prompting-based methods across several VQA benchmarks). We find that ViperGPT's reported gains over BLIP-2 can be attributed to its selection of task-specific modules, and when we run ViperGPT using a more task-agnostic selection of modules, these gains go away. ViperGPT retains much of its performance if we make prominent alterations to its selection of modules: e.g. removing or retaining only BLIP-2. We also compare ViperGPT against a prompting-based decomposition strategy and find that, on some benchmarks, modular approaches significantly benefit by representing subtasks with natural language, instead of code. Our code is fully available at https://github.com/brown-palm/visual-question-decomposition.

## 1 Introduction

End-to-end neural networks (Li et al., 2023) have been the predominant solution for vision–language tasks, like Visual Question Answering (VQA) (Goyal et al., 2017). However, these methods suffer from a lack of interpretability and generalization capabilities. Instead, modular (or neuro-symbolic) approaches (Andreas et al., 2015; Johnson et al., 2017; Hu et al., 2017; Yi et al., 2018) have been long suggested as effective solutions which address both of these limitations. These methods synthesize symbolic programs that are easily interpretable and can be executed (leveraging distinct image or language processing modules) to solve the task at hand. The most recent such models (Gupta and Kembhavi, 2023; Surís et al., 2023; Subramanian et al., 2023) are training-free: they leverage large language models (LLMs) to generate programs and subsume powerful neural networks as modules. Such approaches demonstrate strong results and outperform end-to-end neural networks on zero-shot vision–language tasks.

These recent modular approaches typically include state-of-the-art end-to-end networks, among a complex schema of other modules and engineering designs. As a result, the contribution of these networks is difficult to disentangle from the modularity of their overall system. Thus, in this paper, we analyze ViperGPT, in which BLIP-2 (Li et al., 2023) is a constituent module, as a representative example of a recent and performant modular system for vision–language tasks. BLIP-2 can particularly (and in contrast from ViperGPT's other modules) solve VQA tasks on its own. We ask: where does its additional performance come from, and how much is due to the underlying BLIP-2 model vs. the additional symbolic components? To answer our research questions, we conduct a controlled study, comparing end-to-end, modular, and prompting-based approaches (Sec. 2) on several VQA benchmarks (Sec. 3). We make the following specific contributions:

1. In Section 4, we find that ViperGPT's advantage over BLIP-2 alone is likely due to the task-specific engineering in ViperGPT. Specifically, we run ViperGPT in a task-agnostic setting, in which we do not preselect different subsets of modules for each task (as is done in Surís et al. (2023)). We find that, without the task-specific module selection, the average gain of ViperGPT over BLIP-2 disappears (dropping from +8.7% to -0.8%). Moreover, we find that removing the BLIP-2 module re-

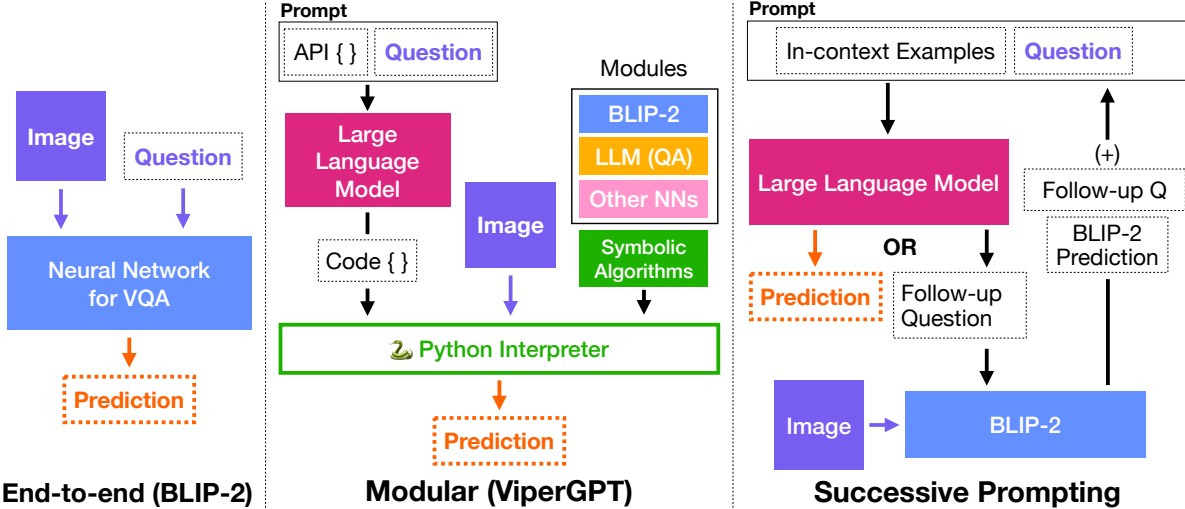

Figure 1: A diagram of the end-to-end, modular, and prompting-based models (Sec. 2) we explore in this paper. Each setting receives an image and question as input and produces a prediction as output (at which time it will terminate). Similar colors across models in this diagram indeed refer to the same modules.

tains a significant percentage of ViperGPT's task-agnostic performance (i.e. 84% for the direct answer setting and 87% for the multiple choice setting). And, retaining only the BLIP-2 module comprises 95% and 122% of ViperGPT's task-agnostic performance in those settings.

2. In Section 5, we find that a prompting-based (rather than code-based) method for question decomposition still constitutes 92% of the performance of an equivalent ViperGPT variant for direct answer benchmarks. Moreover, this method actually exceeds ViperGPT by +12% on average for multiple choice benchmarks. (To the best of our knowledge, our prompting-based method also presents the highest score in the multiple choice setting of A-OKVQA (Schwenk et al., 2022) compared to any other training-free method.) These results suggest that, on some benchmarks, modular approaches significantly benefit by representing subtasks with natural language, instead of code.

3. In Section 6, we explore ViperGPT's generalization to out-of-domain benchmarks. Unlike for in-domain datasets, we find that providing task-specific in-context examples actually leads to a performance drop by 11% for A-OKVQA's direct answer setting and 2% on average for A-OKVQA and ScienceQA multiple choice settings. We additionally ana-

lyze the code that is generated by ViperGPT and observe higher runtime error rates for A-OKVQA's direct answer setting (3%) and the multiple choice benchmarks (12–18%) than the in-domain direct answer benchmarks (1–2%). Finally, while the syntax error rate is 0% for all direct answer benchmarks, it is 1–3% for multiple choice benchmarks.

## 2 Models

In this section, we share specific design decisions and implementation details for each of the three model families we assess in this paper (i.e. end-to-end, modular, and prompting-based). We additionally visualize these approaches in Fig. 1.

### 2.1 End-to-end

As the end-to-end model in our analyses, we use BLIP-2 (Li et al., 2023), an open-source state-of-the-art vision-language model which can be used for image captioning and zero-shot VQA. For VQA, this model first encodes images using a pre-trained image encoder and projects the resulting encoding into the input space of a pre-trained language model. That language model is then used to generate a textual prediction, given the aforementioned image projection and VQA question.

As in Li et al. (2023), we prompt this model with "Question: {} Short answer: []". For the direct answer setting, we generate text directly from the language model. For the multiple choice setting, we select the choice with the maximum log likelihood for text generation.

We use the same settings as Surís et al. (2023, ViperGPT) (i.e. the modular approach in Sec. 2.2) to load and run the model. Specifically, we use the ViT-g/14 image encoder from EVA-CLIP (Fang et al., 2023) and FlanT5-XXL encoder–decoder language model (Chung et al., 2022). We make predictions using 8-bit inference (Dettmers et al., 2022) and generate text with beam search (width = 5, length penalty = -1).

## 2.2 Modular

In this paper, we use ViperGPT (Surís et al., 2023), which is a recent modular system for vision–language tasks.

ViperGPT prompts a language model with a VQA question and an API—which is an interface for manipulating images—to generate a Python program. This API is written in code and describes a `class ImagePatch` and several functions, like `.find`, `.simple_query`, etc. These functions invoke both symbolic algorithms (e.g. for iterating through elements, sorting lists, computing Euclidean distances, etc.) and trained neural network modules (for object detection, VQA, LLM queries, etc.). When the Python program is executed, these functions should manipulate the VQA image and answer the question.

As a simple example, the question "How many black cats are in the image?" might be written as:

```python
def execute_command(image) -> str:
    image_patch = ImagePatch(image)
    cat_patches = image_patch.find('cat')
    black_cat_patches = [
        p for p in cat_patches if
        p.verify_property('cat', 'black')
    ]
    return len(black_cat_patches)
```

The program can then be executed using the Python interpreter and several modules. In this case, `ImagePatch.find(object_name: str) -> list[ImagePatch]` utilizes an object detection module to find all cats and `ImagePatch.verify_property(object_name: str, property: str) -> bool` utilizes text–image similarity for determining whether those cats are black.

**Implementation.** Our implementation of ViperGPT uses the code[1] released with the original

ViperGPT paper (Surís et al., 2023). However, as that codebase currently[2] differs from the original paper in several ways, we have modified it to re-align it with the original report. Specifically, we switch the code-generation model from ChatGPT to Codex, revert the module set and prompt text to those in Surís et al. (2023, Appendix A–B), and otherwise make minor corrections to the behavior and execution of `ImagePatch` and `execute_command`. However, we find that only the full API prompt is made available—not the task-specific prompts—preventing us from exactly replicating Surís et al. (2023).

**Design choices.** In our experiments and like Surís et al. (2023), we prompt Codex (`code-davinci-002`) (Chen et al., 2021a) for code generation and use the same set of neural modules: GLIP (Li et al., 2022), MiDaS (Ranftl et al., 2020), BLIP-2 (Li et al., 2023), X-VLM (Zeng et al., 2022), and InstructGPT (`text-davinci-003`) (Ouyang et al., 2022).

For fairness in comparing with the model in Sec. 2.3, we make a few additional design decisions that deviate from Surís et al. (2023). For our task-agnostic variant (Sec. 4), we use the full `ImagePatch` API and external functions (excluding the `VideoSegment` class) in our prompt. We specify further modifications for our other variants in Sec. 4. We prompt the model with the following signature: "`def execute_command(image) -> str:`" (i.e. we explicitly add "`-> str`" to better conform to the VQA task). During multiple choice, Surís et al. (2023) provides another argument, `possible_choices`, to the signature. However, we extend this argument with an explicit list of these choices.

For the direct answer setting, we use the text as returned by the program as our predicted answer. This text may be generated by a number of modules (e.g. BLIP-2 or InstructGPT) or as a hardcoded string in the program itself. For the multiple choice setting, the returned text is not guaranteed to match a choice in the provided list. So, we map the returned text to the nearest choice by prompting InstructGPT (`text-davinci-003`) with "`Choices: {} Candidate: {} Most similar choice: []`". We select the choice with the highest log likelihood.

We elaborate further on our design choices and how they make our experiments more fair in Appendix B.

---

[1] https://github.com/cvlab-columbia/viper

[2] As of October 22, 2023.

## 2.3 Successive Prompting

Building programs is not the only way to decompose problems. Recent work in NLP has found that large language models improve performance at reasoning tasks when solving problems step-by-step (Wei et al., 2022; Kojima et al., 2022). For question answering, decomposing a question and answering one sub-question at a time leads to further improvements (Press et al., 2022; Zhou et al., 2023; Dua et al., 2022; Khot et al., 2023). Moreover, recent work has started to invoke vision–langauge models based on the outputs of langauge models.

As a convergence of these directions, we introduce a training-free method that jointly and successively prompts an LLM (InstructGPT: `text-davinci-002`) and VLM (BLIP-2) to decompose visual questions in natural language. We call this "Successive Prompting" (following Dua et al. (2022)). At each step, our method uses the LLM to ask one follow-up question at a time. Each follow-up question is answered independently by the vision–language model. In the subsequent step, the LLM uses all prior follow-up questions and answers to generate the next follow-up question. After some number of steps (as decided by the LLM), the LLM should stop proposing follow-up questions and will instead provide an answer to the original question. We constrain the LLM's behavior by appending the more likely prefix of "Follow-up:" or "Answer to the original question:" (i.e. the stopping criteria) to the prompt at the end of each step.

In order to prompt a large language model for this task, we provide a high-level instruction along with three dataset-specific in-context demonstrations of visual question decompositions.

Our method generates text directly, which can be used for the direct answer setting. Like with ViperGPT, we also prompt our method with an explicit list of choices during the multiple choice setting. And, for the multiple choice setting, we select the choice with the highest log likelihood as the predicted answer.

## 3 Evaluation

We evaluate variants of *end-to-end*, *modular*, and *prompt-based* methods (Sec. 2) on a set of VQA benchmarks (Sec. 3.1) using direct answer and multiple choice evaluation metrics (Secs. 3.2 and 3.3).

## 3.1 Benchmarks

We evaluate methods on a set of five diverse VQA benchmarks: VQAv2 (Goyal et al., 2017), GQA (Hudson and Manning, 2019), OK-VQA (Marino et al., 2019), A-OKVQA (Schwenk et al., 2022), and ScienceQA (Lu et al., 2022). We use the following dataset splits as our benchmarks: validation (1000 random samples) for VQAv2, test-dev for GQA, testing for OK-VQA, validation for A-OKVQA, and validation (IMG subset, $QC_IM \rightarrow A$ format) for ScienceQA.

These datasets vary in the amount of perception, compositionality, knowledge, and reasoning their problems require. More specifically: VQAv2 is a longstanding benchmark whose questions require primitive computer vision skills (e.g. classification, counting, etc). GQA focuses on compositional questions and various reasoning skills. OK-VQA requires "outside knowledge" about many categories of objects and usually entails detecting an object and asking for knowledge about that object. A-OKVQA features "open-domain" questions that might also require some kind of commonsense, visual, or physical reasoning. ScienceQA features scientific questions (of elementary through high school difficulty) that require both background knowledge and multiple steps of reasoning to solve. We elaborate further in Appendix D.

## 3.2 Metrics: Direct Answer

We evaluate the *direct answer* setting for VQAv2, GQA, OK-VQA, and A-OKVQA. In this setting, a method will predict a textual answer given an image and question. We report scores using (1) the existing metric for each dataset and (2) the new InstructGPT-eval metric from (Kamalloo et al., 2023).

We observe that while the general trends (determining which models perform better or worse) remain the same between metrics (1) and (2), the actual gap may differ significantly. See Appendix A for further discussion of why (2) is a more robust measure of model performance for our experiments. We include (1) for posterity in Tables 1 and 2, but make comparisons in our text using (2), unless specified otherwise.

**(1) Existing metrics.** GQA uses exact-match accuracy with a single ground truth annotation. VQAv2, OK-VQA, and A-OKVQA use the VQAv2 evaluation metric: a prediction is matched with 10 ground truth annotations (i.e. acc =

`max(1.0, num_matches / 3))`. Note: VQAv2 and OK-VQA pre-process answers before matching.

**(2) InstructGPT-eval.** Kamalloo et al. (2023) find that lexical matching metrics for open-domain question answering tasks in NLP perform poorly for predictions generated by large language models. We make similar observations for the existing direct answer metrics in the VQA datasets we benchmark: such scores correlate quite poorly with our intuitions for open-ended text generated by language models. For example, the prediction "riding a horse" would be marked incorrect when ground truth answers are variants of "horseback riding". Instead, Kamalloo et al. (2023) suggest an evaluation metric (InstructGPT-eval) that prompts InstructGPT (`text-davinci-003`) (Ouyang et al., 2022) with[3]:

```
Question: What is he doing?
Answer: horseback riding
Candidate: riding a horse
Is the candidate correct? [yes/no]
```

Kamalloo et al. (2023) demonstrates a substantial increase of +0.52 in Kendall's $\tau$ correlation with human judgements using their introduced metric instead of exact match on the NQ-Open benchmark (Lee et al., 2019).

### 3.3 Metrics: Multiple Choice

We evaluate the *multiple choice* setting for A-OKVQA and ScienceQA. In this setting, a method will similarly be given an image and question, but also a list of textual choices. The method is required to select one of those choices as its predicted answer. We evaluate this setting using the standard accuracy metric.

## 4 Selecting Modules in ViperGPT

In Surís et al. (2023), the choice of modules is different for each task. This contrasts with end-to-end models like BLIP-2 which are purported to be task-agnostic. To draw a more direct comparison, we evaluate ViperGPT's performance when given the full API (Surís et al., 2023, Appendix B) and set of all corresponding modules. We refer to this as the "task-agnostic" setting (Table 1). We find that, in this case, the gain of ViperGPT over

BLIP-2 is reduced from +6.2% to +2.1% on GQA and +11.1% to -3.6% on OK-VQA (using the existing metrics).[4] We continue to observe that our task-agnostic ViperGPT variant usually does not perform better than BLIP-2 across benchmarks and metrics, with the exception of the multiple choice setting of A-OKVQA, on which ViperGPT does outperform BLIP-2 significantly.

Since ViperGPT relies on BLIP-2 as one of its modules (i.e. in `simple_query` for simple visual queries), we wonder how much influence BLIP-2 has in the ViperGPT framework. Moreover, how much does ViperGPT gain from having modules and functions in addition to BLIP-2?

Accordingly, we run two ablations: we evaluate the performance of ViperGPT without BLIP-2 and with only BLIP-2 (i.e. with no other modules). We also report these evaluations in Table 1.

To do so, we modify the full API prompt provided to ViperGPT. For "only BLIP-2", we delete all modules and functions in the prompt besides `ImagePatch.simple_query`. As the prompt for this module included in-context demonstrations relying on other (now removed) modules, we had to re-write these demonstrations. We either rewrite the existing problem ("zero-shot") or rewrite three random training set examples for each dataset ("few-shot"). For "without BLIP-2", we simply delete `ImagePatch.simple_query` and all references to it from the prompt. We show examples for both procedures in Appendix C.

Because ViperGPT has no other image-to-text modules, we expect that excluding BLIP-2 (i.e. "without BLIP-2") should have a highly detrimental effect on the VQA performance of ViperGPT. However, we instead observe that the variant retains 84% and 87% of the average performance, respectively, for the direct answer and multiple choice benchmarks. This indicates that including many modules improves the robustness of the ViperGPT model, in that ViperGPT is able to compensate by using other modules to replace BLIP-2.

We find that using Viper with BLIP-2 as the only module (i.e. "only BLIP-2") also retains significant performance in the direct answer setting: i.e. by 95% on average. Moreover, this variant actually *gains* performance (+6% on A-OKVQA and +12% on ScienceQA) in the multiple choice setting. This

---

[3]A list of `or`-separated answers is provided when several ground truth annotations are available. Although (Kamalloo et al., 2023) tests if the generated response starts with "yes" or "no", we instead compare the log likelihood of generating "yes" or "no" for additional robustness.

[4]In this comparison, we use our own replicated result for the performance of BLIP-2 on GQA and OK-VQA for consistency, although they may deviate from prior reports (Li et al., 2023) by 2–5%.

| Method | Direct Answer | | | | | | | | Multiple Choice | |
| | VQAv2 | | GQA | | OK-VQA | | A-OKVQA | | A-OKVQA | ScienceQA |
| --- | --- | --- | --- | --- | --- | --- | --- | --- | --- | --- |
| BLIP-2 | 60.8 | 64.8 | 41.9 | 51.1 | 40.8 | 59.5 | 31.8 | 63.5 | 26.2 | 37.0 |
| ViperGPT (task-agnostic) | 62.1 | 63.9 | 44.0 | 52.6 | 37.2 | 57.1 | 39.5 | 61.7 | 52.9 | 36.0 |
| - without BLIP-2 | 45.0 | 55.0 | 31.8 | 46.9 | 7.7 | 49.6 | 4.9 | 44.9 | 43.1 | 33.2 |
| - only BLIP-2 (zero-shot) | 59.3 | 61.0 | 34.7 | 45.5 | 36.0 | 53.0 | 39.3 | 57.8 | 58.9 | 47.7 |
| - only BLIP-2 (few-shot) | 62.9 | 64.4 | 39.0 | 47.8 | 36.4 | 53.8 | 29.9 | 47.2 | 56.6 | 46.6 |
| BLIP-2 (Li et al., 2023) | — | — | 44.7 | — | 45.9 | — | — | — | — | — |
| ViperGPT (Surís et al., 2023) | — | — | 48.1 | — | 51.9 | — | — | — | — | — |

Table 1: Our evaluation of BLIP-2 and our ViperGPT variants across VQA benchmarks. For each direct answer entry, we list both existing metrics (left) and the InstructGPT-eval metric (right) described in Sec. 3.2. As explained in Appendix A, we only include existing metrics for posterity and make comparisons in our text using the InstructGPT-eval metric. We run BLIP-2 using the same inference settings as Surís et al. (2023), which differ slightly from Li et al. (2023).

result seems to indicate that the BLIP-2 module is doing most of the heavy-lifting within ViperGPT for the VQA benchmarks.

## 5  Decomposing Problems with Programs or Prompting?

One of ViperGPT's major contributions is that it decomposes problems into Python programs: it inherently gains the compositionality and logical reasoning that is built into programming languages. However, recent work in NLP suggests that questions can also be iteratively decomposed and solved more effectively than end-to-end approaches using step-by-step natural langauge prompting (Press et al., 2022). Here, we measure the gains related to ViperGPT's choice of building logical, executable programs in Python, rather than by using the interface of natural language and reasoning implicitly within LLMs.

We want to enable as direct a comparison as possible between natural language prompting with our method and program generation with Viper. Thus, we choose the same VLM (BLIP-2) as ViperGPT and an analogous LLM to Codex (`code-davinci-002`)—specifically, we use InstructGPT (`text-davinci-002`). We present the results of our method ("Successive Prompting") in Table 2 and Fig. 2 and directly compare against the "only BLIP-2" variants of ViperGPT. We have also used the same in-context examples for each dataset for both ViperGPT ("only BLIP-2, few-shot") and Successive Prompting, which helps keep the comparison more fair.

Our prompting method performs comparably (i.e. retaining 92% of the performance on average)

to ViperGPT on GQA, OK-VQA, and A-OKVQA in the direct answer setting, and is noticably better (i.e. +4% and +17%) on the multiple choice setting for A-OKVQA and ScienceQA. To the best of our knowledge, our method actually presents the highest A-OKVQA multiple-choice score compared to any other training-free method.

Our method presents intermediate results that are in the form of natural language expressions and strictly subject to downstream operations by neural networks. On the other hand, ViperGPT can present Pythonic data types, like lists and numbers, as well as image regions. Unlike our prompting method, ViperGPT does result in a more diverse set of intermediate representations, some of which can be symbolically manipulated, and is designed to leverage a diverse set of neural networks.

But from this experiment, we determine that it is not strictly necessary to decompose problems using programs in order to realize performance gains. Instead, natural language prompting can offer a simpler alternative. While ViperGPT leverages the intrinsic compositonality and logical execution of programming languages, our method uses conditional generation on intermediate results and a flexible natural language interface for reasoning, while remaining similarly effective.

In Appendix G, we tried to identify patterns in questions to determine whether they were more suitable for formal or natural language-based decomposition. We could not find any clear patterns, following simple question type breakdowns of the original datasets, but are hopeful that future work will explore this further and reveal better insights.

| Method | Direct Answer | | | | | | | | Multiple Choice | |
|---|---|---|---|---|---|---|---|---|---|---|
| | VQAv2 | | GQA | | OK-VQA | | A-OKVQA | | A-OKVQA | ScienceQA |
| ViperGPT (only BLIP-2) | 62.9 | 64.4 | 39.0 | 47.8 | 36.4 | 53.8 | 39.3 | 57.8 | 58.9 | 47.7 |
| Successive Prompting | 53.9 | 57.8 | 37.1 | 47.7 | 28.5 | 45.1 | 36.3 | 55.2 | 63.0 | 64.6 |

Table 2: We evaluate our prompting-based decomposition strategy ("Successive Prompting") from Sec. 5 and compare against the analogous ViperGPT variant ("only BLIP-2") from Sec. 4 and Table 1: both use InstructGPT for decomposition and keep BLIP-2 as the only module. We list the higher score between the zero-shot and few-shot "only BLIP-2" variants here. For each direct answer entry, we list both existing metrics (left) and the InstructGPT-eval metric (right) described in Sec. 3.2.

| | Direct Answer | | | | Multiple Choice | |
|---|---|---|---|---|---|---|
| | VQA | GQA | OK-VQA | A-OKVQA | A-OKVQA | ScienceQA |
| No Exception | 99% | 98% | 99% | 96% | 86% | 79% |
| Parsing | 0% | 0% | 0% | 0% | 1% | 3% |
| Runtime | 1% | 2% | 1% | 4% | 12% | 18% |

Table 3: A breakdown of failure rates across exception modes for ViperGPT (task-agnostic) across our benchmarks. "No Exception" only indicates completion, not correctness, of executions. Parsing errors occur as SyntaxError in Python. We further breakdown runtime errors in Appendix F.

# 6 How well does ViperGPT generalize to out-of-distribution tasks?

As we have observed in Sec. 4, ViperGPT has been designed around a specific set of tasks (including GQA and OK-VQA), especially in its selection of modules and prompt. On the other hand, a core motivation for modular and neuro-symbolic approaches is that these should have better generalization capabilities to unseen tasks. So, we further wonder how robust ViperGPT is to out-of-distribution tasks. In particular, we consider A-OKVQA and (especially) ScienceQA as out-of-distribution (compared to GQA and OK-VQA).

First, we investigate changes to the prompt of ViperGPT. Will adding task-specific in-context examples improve the model's robustness to new tasks? In Table 1, we compare zero-shot and few-shot variants of "ViperGPT (only BLIP-2)". We can see that including few-shot examples consistently improves performance on the "in-domain" tasks (VQAv2, GQA, and OK-VQA) by +2% on average. But, this consistently hurts performance on the "out-of-distribution" tasks (A-OKVQA and ScienceQA) by 11% on A-OKVQA's direct answer setting and 2% on average for their multiple choice settings.

We also look at the correctness of the programs generated by ViperGPT in Table 3. We find that the generated code is (on average) 3x as likely to encounter runtime errors for A-OKVQA compared to the other benchmarks in the direct answer setting. We find that this rate increases by another 3x (i.e. 12%) for A-OKVQA in the multiple choice setting. And, ScienceQA holds the highest rate of runtime failures overall at 18%. In the multiple choice setting, A-OKVQA and ScienceQA produce code that cannot be parsed (i.e. with syntax errors) 1% and 3% of the time. On the other hand, the rate of parsing exceptions is consistently 0% for benchmarks (including A-OKVQA) in the direct answer setting.

# 7 Related Work

**Visual Question Answering.** Visual Question Answering is a common vision–language task with many variants: in our paper, we benchmark several mainstream VQA datasets (Goyal et al., 2017; Hudson and Manning, 2019; Marino et al., 2019; Schwenk et al., 2022; Lu et al., 2022), requiring a broad range of skills: related to computer vision and perception, compositional understanding, outside and world knowledge, scientific and commonsense reasoning, and more.

**End-to-end models.** End-to-end models are the predominant approach in deep learning. In particular, we consider vision–language models here. We observe that recent, state-of-art vision–language models may rely on large-scale pretraining (Radford et al., 2021; Singh et al., 2022; Yu et al., 2022; Wang et al., 2023; Li et al., 2023), be trained on

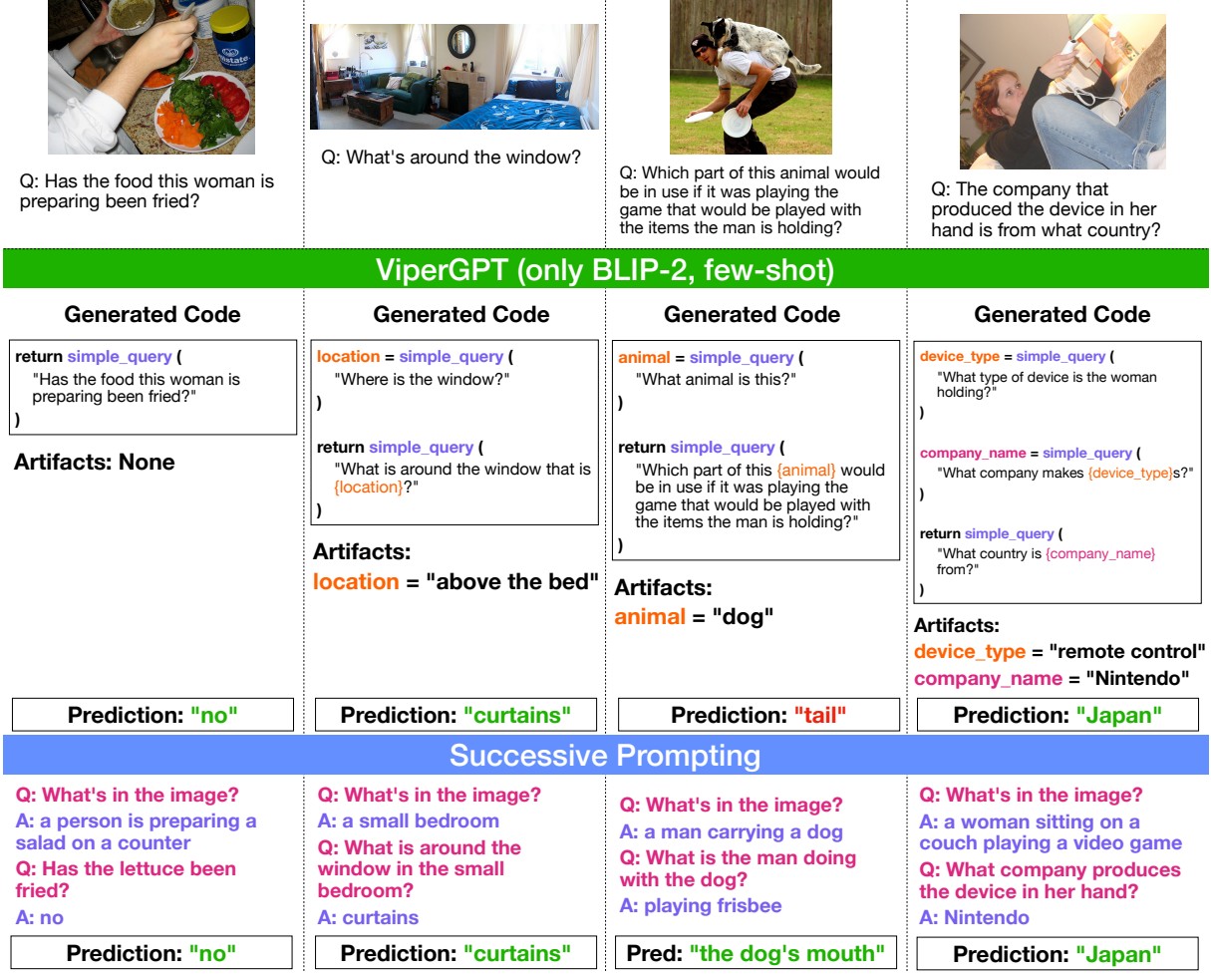

Figure 2: Examples of decompositions for our "ViperGPT (only BLIP, few-shot)" and "Successive Prompting" models on our direct answer benchmarks. These examples have been condensed for readability. In Successive Prompting, follow-up questions (Q) are proposed by the LLM (InstructGPT) and answered (A) by BLIP-2. After a variable number of follow-ups, the LLM will provide a prediction to the original question and terminate.

many tasks with a unified architecture (Chen et al., 2021b; Wang et al., 2022b; Lu et al., 2023), or both (Wang et al., 2022a; Alayrac et al., 2022; Chen et al., 2022). In this paper, we focus on the first category and find BLIP-2 (Li et al., 2023), which utilizes a frozen image encoder and language model, as a suitable and effective representative.

**Neuro-symbolic methods.** Several prior methods have attempted neuro-symbolic approaches for visual reasoning tasks (Andreas et al., 2015; Johnson et al., 2017; Hu et al., 2017; Yi et al., 2018). However, until now, these methods were learned by training and found middling success in doing so. More recently, a few training-free approaches have been suggested that leverage the powerful in-context and program generation abilities of modern large language models (Gupta and Kembhavi,

2023; Surís et al., 2023; Subramanian et al., 2023). Simultaneously, these approaches adopt SOTA neural networks in their set of modules. All together, these methods present the first competitive neuro-symbolic solutions for visual reasoning tasks.

**Natural language reasoning and tool-use.** Recently, it has been found in NLP that large language models can more effectively perform reasoning tasks by reasoning step-by-step (Wei et al., 2022; Kojima et al., 2022). A few works have extended a similar capability in LLMs for complex problem and question decomposition (Press et al., 2022; Zhou et al., 2023; Dua et al., 2022; Khot et al., 2023). Finally, recent works have learned ways to prompt language models to call tools (Parisi et al., 2022; Schick et al., 2023). All of these emergent research directions jointly enable the prompting

approach we present in Sec. 5.

# 8 Conclusion

In this paper, we have analyzed ViperGPT (Surís et al., 2023), a recent and intricate modular approach for vision–language tasks. We unbox ViperGPT, asking the research question: where does its performance come from? Observing that one of ViperGPT's five modules (i.e. BLIP-2 (Li et al., 2023)) often outperforms or constitutes a majority of its own performance, we investigate this module's role further. Through our experiments, we find that while ViperGPT's marginal gains over BLIP-2 are a direct result of its task-specific module selection, its modularity improves its overall robustness and it can perform well even without the (seemingly critical) BLIP-2 module. Additionally, we investigate ViperGPT's choice of generating Python programs. We ask if this is necessary and, alternatively, propose a method relying on prompting large language models and vision–language models to instead decompose visual questions with natural language. Our method performs comparably to the relevant ViperGPT variants and, to the best of our knowledge, even reports the highest multiple choice accuracy on A-OKVQA (Schwenk et al., 2022) by any training-free method to date.

# Limitations

Although the experiments in our paper are self-contained and designed to be directly comparable with each other, the absolute scores we report differ from other reports. For example, our results for BLIP-2 on GQA and OK-VQA are within 2–5% of the original reports. We attribute such differences to possible differences in model inference settings between (Surís et al., 2023)—which we follow and e.g. runs the model with 8-bit inference—and (Li et al., 2023).

In our "ViperGPT (only BLIP-2)" experiments, we find that the method often calls BLIP-2 directly in a single step and might not offer any mechanisms beyond BLIP-2 in that case.

# Acknowledgments

We would like to thank Michael A. Lepori for his generous feedback on this work. This work is partially supported by the Samsung Advanced Institute of Technology and a Brown University Presidential Fellowship for Apoorv Khandelwal. Our research was conducted using computational resources and services at the Center for Computation and Visualization, Brown University.

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

# A Existing Metrics vs. InstructGPT-eval

We observe that existing VQA metrics correlate poorly with (our) human judgments for evaluating model predictions. We analyze predictions for our model types (i.e. "BLIP-2", "ViperGPT (task-agnostic)", and "Successive" in Tables 1 and 2) on random training split subsets ($N = 50$) of VQAv2, GQA, and A-OKVQA. We specifically find that, when the existing and new (InstructGPT-eval) evaluation metrics disagree, InstructGPT-eval is correct 93% of the time. In Fig. 3, we show two examples of such disagreements between existing VQA metrics and the open-ended metric. Therefore, we include the results of existing metrics in our paper for posterity, but do not find these reliable (especially for open-ended text generated by models like InstructGPT). We instead make comparisons in our paper using the InstructGPT-eval metric. We observe that trends (i.e. which model performs better) are usually the same for both metrics, but the actual gaps may differ significantly.

# B ViperGPT Design Choices

We make a few modifications (listed in Sec. 2.2) to the ViperGPT method (from the original design (Surís et al., 2023)) to improve conformity for the VQA task and ensure fairness when comparing to our prompting-based approach in Sec. 2.3. We elaborate further here.

As we always expect the executable program to return a string for VQA, we explicitly add "-> str" to the function signature in the prompt. By design, our prompting-based approach can similarly only result in a string.

For the multiple choice setting, we provide an explicit list of choices in the code-generation prompt (e.g. "# possible answers : ['dog', 'cat', 'foo', 'bar']" after the question prompt). We do this, so the code generation model benefits from awareness of these choices when generating the program. Similarly, our prompting-based method is provided with a list of choices in conjunction with the question, prior to proposing follow-up questions.

Unlike the Successive Prompting method, the ViperGPT program is not guaranteed to produce a result that matches one of the multiple choices. So we map this result to the most similar choice using InstructGPT (text-davinci-003). We make this choice because text-davinci-003 is already used by a module in ViperGPT and for the open-ended evaluation metric.

# C ViperGPT Variants

```
>>> # Which kind of animal is not eating?
>>> def execute_command(image) -> str:
>>>    image_patch = ImagePatch(image)
>>>    animal_patches = image_p.find("animal")
>>>    for animal_patch in animal_patches:
>>>      if not animal_patch.verify_property(
           "animal", "eating"
         ):
  >>>        return animal_patch.simple_query(
             "What kind of animal is eating?"
           ) # crop would include eating so
             keep it in the query
>>>    # If no animal is not eating,
       query the image directly
>>>    return image_patch.simple_query(
         "Which kind of animal is not eating?"
       )
```

```
>>> # Which kind of animal is not eating?
>>> def execute_command(image) -> str:
>>>    image_patch = ImagePatch(image)
>>>    animals = image_patch.simple_query("
         Which animals are in the image?
       ")
>>>    return image_patch.simple_query(f"
         Which of the {animals} is not eating?
       ")
```

Figure 4: Partial prompt for ViperGPT (only BLIP-2, zero-shot), showing how the existing demonstration is re-written with only the simple_query module.

```
>>> # Are there both windows and doors in this
    # photograph?
>>> def execute_command(image) -> str:
>>>    image_patch = ImagePatch(image)
>>>    windows_present = image_p.simple_query(
         "Are there windows in this image?"
       )
>>>    doors_present = image_p.simple_query(
         "Are there doors in this image?"
       )
>>>    if windows_present == "yes"
       and doors_present == "yes":
>>>      return "yes"
>>>    else:
>>>      return "no"
```

Figure 5: Partial prompt for ViperGPT (only BLIP-2, few-shot), showing one in-context demonstration written for GQA.

# D Datasets

1. **VQAv2**: This is a classic VQA benchmark. Many of the questions involve tasks (like classification, attribute detection, and counting) and require primitive computer vision skills.

```
Question (VQAv2): What does this sign mean?
Answers: twisty narrow road, narrow road (x4), curves/narrow
    road ahead, ...

Prediction (BLIP-2): warning of a curve ahead
Actual: Correct
Existing metric: "Incorrect"
InstructGPT-eval metric: "Correct"
```

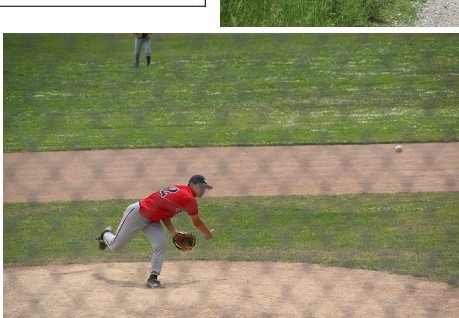

```
Question (A-OKVQA): Why is he bent over?
Answers: pitching stance, follow through (x2),
    pitching (x4), throwing ball, ...

Prediction (BLIP-2): to throw the ball
Actual: Correct
Existing: "Incorrect"
InstructGPT-eval: "Correct"
```

Figure 3: Two examples of disagreements between existing VQA metrics and InstructGPT-eval from Kamalloo et al. (2023). In all, InstructGPT-eval *correctly* evaluates the predictions. As we mention in Sec. 3.2, we find evaluations from InstructGPT-eval more accurate in an overwhelming majority of cases.

|  | Direct Answer | | | | Multiple Choice | |
|---|---|---|---|---|---|---|
|  | VQA | GQA | OK-VQA | A-OKVQA | A-OKVQA | ScienceQA |
| NameError | 100% | 14% | 25% | 13% | 61% | 34% |
| AttributeError | 0% | 42% | 25% | 63% | 22% | 36% |
| IndexError | 0% | 13% | 13% | 6% | 6% | 21% |
| TypeError | 0% | 15% | 19% | 19% | 7% | 4% |
| IndentationError | 0% | 0% | 0% | 0% | 0% | 2% |
| ValueError | 0% | 3% | 6% | 0% | 3% | 1% |
| KeyError | 0% | 0% | 0% | 0% | 0% | 1% |
| ZeroDivisionError | 0% | 0% | 0% | 0% | 0% | 0% |
| Other | 0% | 14% | 13% | 0% | 0% | 0% |

Table 4: Breakdown of failure rates across runtime exceptions for ViperGPT (task-agnostic) across our benchmarks.

For example, "What color are the pants?" might entail (1) detecting the pants in the image and (2) determining their color.

2. **GQA**: This is a benchmark that focuses on compositional questions. Requires an "array of reasoning skills such as object and attribute recognition, transitive relation tracking, spatial reasoning, logical inference and comparisons" (Hudson and Manning, 2019). For example, "What color are the cups to the left of the tray on top of the table?" is a multi-step composition (focusing on spatial relationships and attribute recognition).

3. **OK-VQA**: Requires "outside knowledge"

about many categories of objects. Usually requires detecting an object and asking for knowledge about that object. Example (Surís et al., 2023, Figure 5): "The real live version of this toy does what in the winter?". Involves locating and identifying the toy, then asking about the rest.

4. **A-OKVQA**: A follow-up benchmark to OK-VQA. Instead of asking for closed-domain knowledge about objects, this features "open-domain" questions that might also require some kind of commonsense, visual, or physical reasoning. For example, "Which position will the red jacket most likely finish in?" in-

volves (1) identifying the context (a ski race), (2) locating all the racers, (3) identifying the racer who is wearing the red jacket, (4) determining the orientation of the race (e.g. left-to-right), and (5) determining the "index" of the red jacket racer among all racers along this orientation. The question's textual prior appears contextually insufficient and proposing a program based on this alone (as in ViperGPT) could be fragile and quite difficult.

5. **ScienceQA**: This benchmark features scientific questions (of elementary through high school difficulty) that require both background knowledge and multiple steps of reasoning to solve. Their example question is "Which type of force from the baby's hand opens the cabinet door?" (choices: push, pull). The given reasoning is that (paraphrased) "The direction of push is away from and pull is towards the acting object. The baby's hand applies a force to the cabinet door that causes the door to open. The direction of the door opening is towards the baby, so the force is pull." Without seeing the image, it is not apparent, but what determines whether the baby is opening or closing the door is the fine-grained detail that the baby's hand is curled over the top of the cabinet door, not grasping the handle or pushing the door's surface. The current stage of visual programming models are not capable of such difficult multi-step reasoning chains and planning around such fine details. Instead, we find that ViperGPT tends to default to its end-to-end VQA module instead.

## E  Log likelihood of generating continuations

We use a weighted byte-length normalization for generating the log likelihood of a continuation, i.e.

$$\sum_{j=m}^{m+(n-1)} \log P(x_j|x_{0:j}) \frac{L_{x_k}}{\sum_{k=m}^{m+(n-1)} L_{x_k}} \quad (1)$$

where $x$ is a list of tokens (with $m$ tokens in the prompt and $n$ tokens in the continuation) and the byte-length of the token $x_i$ is $L_{x_i}$.

## F  Runtime Failure Rate of ViperGPT

As an extension to Table 3, we further breakdown ViperGPT failures for runtime errors in Table 4.

## G  Failure Rates By Question Type

| | BLIP-2 | ViperGPT | Successive |
|---|---|---|---|
| **GQA** (15) | | | |
| chooseAttr | 62% | 67% | 49% |
| chooseRel | 65% | 72% | 60% |
| verifyObj | 72% | 80% | 70% |
| queryAttr | 43% | 50% | 40% |
| verifyAttr | 58% | 67% | 59% |
| logicalObj | 62% | 68% | 61% |
| logicalAttr | 59% | 64% | 57% |
| queryGlobal | 29% | 34% | 29% |
| verifyRel | 58% | 61% | 56% |
| queryCat | 50% | 46% | 42% |
| queryRel | 42% | 39% | 39% |
| compareAttr | 56% | 56% | 59% |
| chooseCat | 84% | 53% | 66% |
| **OK-VQA** (11) | | | |
| 10 (Weather) | 65% | 62% | 45% |
| 6 (Social Sci.) | 49% | 55% | 39% |
| 5 (Food) | 63% | 58% | 43% |
| Other | 63% | 58% | 45% |
| 7 (People) | 63% | 60% | 46% |
| 8 (Plants) | 57% | 57% | 44% |
| 2 (Brands) | 53% | 57% | 44% |
| 4 (Sports) | 63% | 62% | 52% |
| 9 (Science) | 54% | 52% | 43% |
| 3 (Objects) | 60% | 56% | 47% |
| 1 (Vehicles) | 54% | 51% | 43% |
| **ScienceQA** (15) | | | |
| us-history | 33% | 47% | 43% |
| earth-science | 51% | 56% | 62% |
| biology | 40% | 56% | 76% |
| chemistry | 42% | 25% | 47% |
| economics | 53% | 3% | 27% |
| physics | 43% | 34% | 58% |
| science-practices | 46% | 21% | 51% |
| geography | 23% | 27% | 75% |

Table 5: We breakdown failure rates for our three model families by question types (as specified in the original GQA, OK-VQA, and ScienceQA datasets). We remove outliers (i.e. categories with $< 50$ samples) and sort by |ViperGPT − Successive|.

## H  Successive Prompting Example

```
Question: Has the food this woman is
    preparing been fried?
Follow-up: What's in the image?
Follow-up answer: a person is preparing
    a salad on the counter
Follow-up: Has the lettuce been fried?
Follow-up answer: no
Answer to the original question: no
```