# OpenReview forum: "Analyzing Modular Approaches for Visual Question Decomposition"
_EMNLP/2023/Conference — EMNLP 2023 Main_

### Official Review · Reviewer_oZRE · 2023-08-04

**Soundness:** 3

**Excitement:**

3: Ambivalent: It has merits (e.g., it reports state-of-the-art results, the idea is nice), but there are key weaknesses (e.g., it describes incremental work), and it can significantly benefit from another round of revision. However, I won't object to accepting it if my co-reviewers champion it.

**Paper Topic And Main Contributions:**

This work compared prompt-based and neural-symbolic methods on several vision-language tasks, and found that the prompt-based method (based on BLIP-2) can achieve comparable results as the neural-symbolic method (ViperGPT).

**Questions For The Authors:**

see weakness.

**Reasons To Accept:**

A number of visual language benchmarks are used an the results support the main claim of the paper. Ablating ViperGPT and find out the main contributor of the improved performance is also valuable.

**Reasons To Reject:**

My main concern is that in the area of language model, it is widely aware that simple QA tasks does not need complicated reasoning skills, and some simple benchmarks cannot even differentiate between GPT-3 and GPT-4. I am not sure if the applied benchmarks contain complext reasoning tasks, e.g. math, programming, multi-hop qa, etc. If not, then the paper does not bring new insights - it is already known that a pretrained model, e.g. LLaMA and BLIP-2 can handle questions that is not complicated enough without any neural symbolic add-ons.

I humbly ask the authors to introduce more about the benchmarks and their difficulties. Otherwise the conclusion of the paper is not strong or grounded enough.

**Reproducibility:**

3: Could reproduce the results with some difficulty. The settings of parameters are underspecified or subjectively determined; the training/evaluation data are not widely available.

**Reviewer Confidence:**

2: Willing to defend my evaluation, but it is fairly likely that I missed some details, didn't understand some central points, or can't be sure about the novelty of the work.

---

> ### Author Rebuttal · Authors · 2023-08-29
>
> Thank you for reviewing our paper and for your response!
>
> > I am not sure if the applied benchmarks contain complex reasoning tasks… I humbly ask the authors to introduce more about the benchmarks and their difficulties.
>
> We chose this set of benchmarks because they require diverse skills and are indeed challenging. We summarize them below and are happy to do so further in our paper. Of course, the benchmarks’ original papers provide even more detail (especially showing examples with images).
>
> Finally, we would like to emphasize that these benchmarks remain unsolved by current SOTA models with large pre-training and billions of parameters. One open-source example is BLIP-2, whose paper we can refer to (BLIP-2, Table 1) and which we include in our own analyses (Table 1). And, we can also refer to closed-source models, such as PaLI-17B, which reports a score (PaLI, Xi Chen et al., ICLR 2023, Table 3) that is far from saturated on OK-VQA. These latest results are a testament to the challenging nature of the benchmarks we have selected.
>
> ---
> ---
>
> ## Summary of the benchmarks used in this paper
>
> **VQAv2:** This is a classic/standard VQA benchmark. It would be the “least challenging” among our selection. Many of the questions involve tasks (like classification, attribute detection, and counting) and require primitive computer vision skills, which could be answered in a small number of steps. For example, “What color are the pants?” might entail (1) detecting the pants in the image and (2) determining their color.
>
> **GQA:** This is a benchmark that focuses on compositional questions, which are very suitable for benchmarking modular approaches. Requires an “array of reasoning skills such as object and attribute recognition, transitive relation tracking, spatial reasoning, logical inference and comparisons” (ref. GQA paper). For example, “What color are the cups to the left of the tray on top of the table?” is a multi-step composition (focusing on spatial relationships and attribute recognition).
>
> **OK-VQA:** Requires “outside knowledge” about many categories of objects. Usually requires detecting an object and asking for knowledge about that object. Example (ViperGPT Figure 5): “The real live version of this toy does what in the winter?”. Involves locating and identifying the toy, then asking about the rest.
>
> **A-OKVQA:** A follow-up benchmark to OK-VQA. Instead of asking for closed-domain knowledge about objects, this features “open-domain” questions that might also require some kind of commonsense, visual, or physical reasoning. For example, “Which position will the red jacket most likely finish in?” involves (1) identifying the context (a ski race), (2) locating all the racers, (3) identifying the racer who is wearing the red jacket, (4) determining the orientation of the race (e.g. left-to-right), and (5) determining the “index” of the red jacket racer among all racers along this orientation. We’d like to point out that the question’s textual prior is insufficient and proposing a program based on this alone (as ViperGPT attempts to do) could be fragile and quite difficult!
>
> **ScienceQA:** This benchmark features scientific questions (of elementary through high school difficulty) that require both background knowledge and multiple steps of reasoning to solve. Their example question is “Which type of force from the baby’s hand opens the cabinet door?” (choices: push, pull). The given reasoning is that (paraphrased) “The direction of push is away from and pull is towards the acting object. The baby’s hand applies a force to the cabinet door that causes the door to open. The direction of the door opening is towards the baby, so the force is pull.” Without seeing the image, it is not apparent, but what determines whether the baby is opening or closing the door is the fine-grained detail that the baby’s hand is curled over the top of the cabinet door, not grasping the handle or pushing the door’s surface. The current stage of visual programming models are not capable of such difficult multi-step reasoning chains and planning around such fine details. Instead, we find that ViperGPT tends to default to its end-to-end VQA module instead.
>
> Note: GQA and OK-VQA were benchmarked in the original ViperGPT paper (please see ViperGPT Figures 4-5).

---

### Official Review · Reviewer_a58D · 2023-08-05

**Soundness:** 3

**Excitement:**

4: Strong: This paper deepens the understanding of some phenomenon or lowers the barriers to an existing research direction.

**Paper Topic And Main Contributions:**

First, This study analyzes ViperGPT's performance gain to explore the power of question decomposition in visual question answering tasks. Then, the authors propose an iterative prompting-based approach for visual question decomposition to achieve similar performance. The proposed method is different from the neuro-symbolic approach that, without the logical and executable program, the large language model can also decompose and tackle complex visual question answering problems.

**Questions For The Authors:**

The study is ViperGPT-specific. How is it translatable to other models (and even other tasks)?

**Reasons To Accept:**

- The study explores the power of the neuro-symbolic approach in visual question answering and proposes an strong alternative, revealing the root of the benefit of question decomposition.
- Experimental results are quite supportive.

**Reasons To Reject:**

The manuscript's organization confuses the reader of the purpose of the research. The first part of the study is like an extension of ViperGPT's ablation study, making it less attractive.

**Reproducibility:**

4: Could mostly reproduce the results, but there may be some variation because of sample variance or minor variations in their interpretation of the protocol or method.

**Reviewer Confidence:**

4: Quite sure. I tried to check the important points carefully. It's unlikely, though conceivable, that I missed something that should affect my ratings.

**Typos Grammar Style And Presentation Improvements:**

The manuscript's organization prevents the delivery of the study's core findings. Re-organizing to highlight and focus on the power of decomposition might be more interesting to read.

---

> ### Author Rebuttal · Authors · 2023-08-29
>
> Thanks for your efforts in reviewing our paper! We’d like to answer your questions / concerns directly:
>
> > The first part of the study is like an extension of ViperGPT’s ablation study, making it less attractive. Re-organizing to highlight the power of decomposition might be more interesting.
>
> We would like to clarify that our paper is not an extension of ViperGPT’s ablations, rather a carefully controlled set of experiments that serve as both a reproducibility study and a new set of ablations and hypotheses. In the process, we also propose a natural language-based decomposition strategy as a similarly performant alternative to program generation.
>
> We do appreciate your advice about the organization of our paper and would like to propose some adjustments. We will:
>  - List the main findings in the Introduction.
>  - Add a diagram to Section 2, illustrating the differences between end-to-end, neuro-symbolic, and prompting-based methods.
>  - Include qualitative/visual results of each method with our experiments.
>  - List the research questions in a new subsection 2.1.
>  - Highlight the relevant research questions at the beginning of each experimental section.
>
> We would appreciate your feedback on whether these would address your concerns (or if not, what concerns remain). We will be happy to make the agreed changes for camera-ready.
>
> > This study is ViperGPT-specific. How is it translatable to other models (and even other tasks)?
>
> It’s true that our experiments are specific to ViperGPT and VQA-style benchmarks. We did this to prioritize depth over breadth and feature experiments most suited for ViperGPT and our proposed prompting method. We are very interested in analyzing other methods and tasks and better understanding how general these results are. This would entail significant model-specific experimental design in each case, so it is something that we cannot realistically fit into this paper, but do hope to pursue in follow-up work!

---

### Official Review · Reviewer_9Ggp · 2023-08-06

**Soundness:** 4

**Excitement:**

4: Strong: This paper deepens the understanding of some phenomenon or lowers the barriers to an existing research direction.

**Paper Topic And Main Contributions:**

The paper provides a thorough analysis of the recent works on visual programs, primarily focusing on ViperGPT. The work conducts experiments across 5 VQA datasets and compares end-to-end and visual programming approaches. The main discoveries are:
1. BLIP-2 is disproportionately influential in ViperGPT.
2. ViperGPT can get similar performance by using natural language question decomposition rather than Python programs.

**Questions For The Authors:**

Error analysis of ViperGPT. It would be interesting to see where the errors come from. For example, the program might be not runnable; the program might be unreasonable; or one component is giving a wrong answer.  Maybe investigate 100 examples and count how many errors each category has.

**Reasons To Accept:**

The paper provides great insight into the recent visual programming framework. This area has been really popular, especially after Visprog won CVPR best paper. However, there lack of work that analyzes where is the performance gain coming from.

The discoveries in this paper are inspiring and provoke more thinking. It seems that ViperGPT is just a shallow wrapper of BLIP-2. The ablation studies are complete and interesting. The NLP community will be benefited from the message this paper shows.

**Reasons To Reject:**

1. The performance gain of visual programming might be really dependent on the task. For example, ViperGPT may have a bigger performance gain on counting tasks like TallyQA. The paper would be more interesting if it provides some analysis of the differences between the tasks, and how they influence the gain of visual programming compared with BLIP-2.

2. There could be more analysis on when ViperGPT makes errors. Neuro-symbolic approaches are typically more fragile. For example, the program maybe simply not runnable and has execution errors. It would be great if more error analysis is made in this paper.

**Reproducibility:**

4: Could mostly reproduce the results, but there may be some variation because of sample variance or minor variations in their interpretation of the protocol or method.

**Reviewer Confidence:**

4: Quite sure. I tried to check the important points carefully. It's unlikely, though conceivable, that I missed something that should affect my ratings.

**Typos Grammar Style And Presentation Improvements:**

There are minor typos (e.g. "langauge" on line 190, 191, 414). Some proofreading could be helpful.

---

> ### Author Rebuttal · Authors · 2023-08-29
>
> Thank you very much for your time and consideration! We really value your feedback and agree that these changes will greatly strengthen the paper. Please see how we propose to address your points below.
>
> > The paper would be more interesting if it provides some analysis of the differences between the tasks, and how they influence the gain of visual programming compared with BLIP-2.
>
> We would be happy to elaborate further about each of our benchmark tasks in revision.
>
> We’ve measured the failure rates of BLIP-2 / ViperGPT across each benchmark’s question types (sorted by ViperGPT’s improvement over BLIP-2). Please refer to the **Breakdown by Question Type** section below for the details. For space considerations, we have shared the bottom/top 3 entries there (and will include the complete results in revision, including the breakdown for our Prompting method as well as associated qualitative/visual examples).
>
> Generally, we find that ViperGPT performs better than BLIP-2 at visual recognition / counting tasks due to its vision modules, but worse on certain (module-dependent) skills (e.g. diagram understanding or text recognition) and when generating programs for questions that require complex reasoning or are themselves image-dependent.
>
> > ViperGPT may have a bigger performance gain on counting tasks like TallyQA.
>
> Counting is actually one category of VQAv2 and corresponds with the following row: as we expect, Viper does improve compared to BLIP-2.
>
> |          | BLIP-2 | Viper |
> |----------|--------|-------|
> | how many |   0.36 |  0.47 |
>
> > There could be more analysis on when ViperGPT makes errors.
>
> You are indeed correct that program-based approaches can be fragile and fail upon parsing or runtime. This is a great point in favor of program-free methods, such as prompting with natural language. We have measured the error-rate along your proposed breakdown over all of ViperGPT’s mistakes: please refer to the **Breakdown for ViperGPT Errors** section below for the details. We will also include qualitative examples in our revision.
>
> We actually find that the ViperGPT method encounters few parsing and runtime errors for benchmarks that it was designed for (i.e. GQA, OK-VQA — and similarly, VQAv2 and A-OKVQA). We observe a significant increase in the rates of these errors for the A-OKVQA MC subset (which contains additional, ambiguously worded questions that require choices to contextualize correctly) and very different benchmarks, like ScienceQA. Such fragility is detrimental to the overall performance in these out-of-domain tasks, leaving up to 15-20% accuracy on the table.
>
> # Additional details (if interested)
>
> ---
> ---
>
> ## Breakdown by Question Type
>
> You may also find further details about these question types in each benchmark’s original papers (i.e. VQAv2: Figure 4, GQA: Appendix Table 2, OK-VQA: Figure 4, and ScienceQA: Figure 4). Note: A-OKVQA is categorized by reasoning type (knowledge-based, commonsense, visual, physical) but provides no such annotations in its released dataset.
>
>
> ### VQAv2
>
> We can see that ViperGPT performs better on questions such as “are there” (involving object detection and counting) as compared to  “do” or “why” (which require causal reasoning).
>
> |               | BLIP-2 | Viper    |
> |-----------|--------|----------|
> | why       | 0.50   | 0.25     |
> | do        | 0.75   | 0.50     |
> | what are  | 0.83   | 0.58     |
> | ...       |        |          |
> | do you    | 0.80   | 1.00     |
> | who is    | 0.40   | 0.60     |
> | are there | 0.61   | 0.83     |
>
>
> ### GQA
>
> ViperGPT performs better on “queryAttr” (i.e. What color is the apple?) and “exist” (i.e. Is there an apple?) than “verifyGlobal” (i.e. Is it cloudy today?).
>
> |                    | BLIP-2 | Viper    |
> |--------------|--------|----------|
> | chooseObject |   0.88 |     0.56 |
> | verifyGlobal |   0.84 |     0.69 |
> | chooseRel    |   0.75 |     0.60 |
> | …            |        |          |
> | exist        |   0.71 |     0.79 |
> | queryAttr    |   0.40 |     0.49 |
> | queryGlobal  |   0.19 |     0.31 |
>
>
> ### OK-VQA
>
> Questions are generally easily decomposable, regardless of the given question types. However, “Brands, Companies & Products” typically require reading logos or text (to identify the brand/product), which certain modules may struggle with. Unlike the end-to-end model, the decomposition approach is very strongly dependent on detecting this attribute correctly.
>
> |                                                           | BLIP-2 | Viper |
> |------------------------------------------|--------|-------|
> | Brands, Companies & Products             |   0.70 |  0.59 |
> | Weather & Climate                        |   0.78 |  0.67 |
> | Vehicles & Transportation                |   0.57 |  0.49 |
> | …                                        |        |       |
> | Plants & Animals                         |   0.55 |  0.54 |
> | Sports & Recreation                      |   0.58 |  0.59 |
> | Geography, History, Language and Culture |   0.35 |  0.54 |
>
>
> ### ScienceQA
>
> ViperGPT is best at the “word-study” topic (i.e. “Which term matches the picture?” for a given list of choices). This can be very directly written as a 1-step program that uses Viper’s text-image similarity module between the image and choices. On the other hand, the “economics” task depends on whether two individuals can “trade” items based on parsing a diagram with either natural images or text (to determine whether the items are available). Such diagram-based questions are probably well outside the domain of questions that ViperGPT was designed for and generating a task-specific program would be very difficult without further inductive biases.
>
> |                                | BLIP-2 | Viper |
> |-----------------------|--------|-------|
> | civics                |   1.00 |  0.00 |
> | economics             |   0.53 |  0.03 |
> | literacy-in-science   |   0.50 |  0.17 |
> | …                     |        |       |
> | vocabulary            |   0.20 |  0.40 |
> | reading-comprehension |   0.25 |  0.63 |
> | word-study            |   0.20 |  0.60 |
>
> ---
> ---
>
> ## Breakdown for ViperGPT Errors
>
> |                       | VQAv2 | GQA | OK-VQA | A-OKVQA | A-OKVQA (MC) | ScienceQA (MC) |
> |-----------------------|-------|-----|--------|---------|--------------|----------------|
> | Total Incorrect       |   361 | 599 |    439 |     412 |          540 |           1283 |
> | No Exception          |   99% | 99% |    99% |     96% |          86% |            79% |
> | Parsing (SyntaxError) |    0% |  0% |     0% |      0% |           1% |             3% |
> | Runtime               |    1% |  1% |     1% |      4% |          12% |            18% |
>
> We can specifically breakdown the runtime exceptions as well:
>
> |                  | VQAv2 | GQA | OK-VQA | A-OKVQA | A-OKVQA (MC) | ScienceQA (MC) |
> |------------------|-------|-----|--------|---------|--------------|----------------|
> | NameError        |  100% |  0% |    20% |     13% |          61% |            34% |
> | AttributeError   |    0% | 33% |    40% |     63% |          22% |            36% |
> | IndexError       |    0% |  0% |     0% |      6% |           6% |            21% |
> | TypeError        |    0% | 67% |    40% |     19% |           7% |             4% |
> | IndentationError |    0% |  0% |     0% |      0% |           0% |             2% |
> | ValueError       |    0% |  0% |     0% |      0% |           3% |             1% |
> | KeyError         |    0% |  0% |     0% |      0% |           0% |             1% |

---

### Meta-Review · Area_Chair_oMYq · 2023-09-23

**Recommendation:** 5

**Metareview:**

This paper conducts a comprehensive analysis of visual programming in the context of visual question answering (VQA), with a particular focus on ViperGPT. The key contributions include highlighting the power of question decomposition in VQA tasks, proposing an iterative prompting-based approach for visual question decomposition, and offering insights into the performance gain achieved by visual programming in ViperGPT. Overall, this paper offers valuable insights into the role of visual programming in VQA and presents an alternative approach to question decomposition. Both the AC and reviewers are excited about this work. With some improvements in organization and additional context in the revision, it can make a significant contribution to the field.

---

### Decision · Program_Chairs · 2023-10-07

**Decision:**

Accept-Main

**Comment:**

This paper conducts a comprehensive analysis of visual programming in the context of visual question answering (VQA), with a particular focus on ViperGPT. The key contributions include highlighting the power of question decomposition in VQA tasks, proposing an iterative prompting-based approach for visual question decomposition, and offering insights into the performance gain achieved by visual programming in ViperGPT. Overall, this paper offers valuable insights into the role of visual programming in VQA and presents an alternative approach to question decomposition. Both the AC and reviewers are excited about this work. With some improvements in organization and additional context in the revision, it can make a significant contribution to the field.